# PixelFade: Privacy-preserving Person Re-identification with Noise-guided Progressive Replacement

## ABSTRACT

Online person re-identification services face privacy breaches from potential data leaks and recovery attacks, exposing cloud-stored images to malicious attackers and triggering public concern. The privacy protection of pedestrian images is crucial. Previous privacy-preserving person re-identification methods are unable to resist recovery attacks and compromise accuracy. In this paper, we propose an iterative method (PixelFade) to optimize pedestrian images into noise-like images to resist recovery attacks. We first give an in-depth study of protected images from previous privacy methods, which reveal that the **chaos** of protected images can disrupt the learning of recovery networks, leading to a decrease in the power of the recovery attacks. Accordingly, we propose Noise-guided Objective Function with the feature constraints of a specific authorization model, optimizing pedestrian images to normal-distributed noise images while preserving their original identity information as per the authorization model. To solve the above non-convex optimization problem, we propose a heuristic optimization algorithm that alternately performs the Constraint Operation and the Partial Replacement operation. This strategy not only safeguards that original pixels are replaced with noises to protect privacy, but also guides the images towards an improved optimization direction to effectively preserve discriminative features. Extensive experiments demonstrate that our PixelFade outperforms previous methods in resisting recovery attacks and Re-ID performance. The code will be released.

## CCS CONCEPTS

• **Computing methodologies** → **Object identification**.

## KEYWORDS

person re-identification, privacy protection, pedestrian images, adversarial attacks

## 1 INTRODUCTION

With the flourishing of deep learning, person re-identification (Re-ID) is widely used in various surveillance systems [28]. Given a query person, the purpose of Re-ID is to match pedestrians appearing under different cameras at a distinct time. This necessitates uploading pedestrian images captured by various cameras to a cloud-based storage system to streamline the Re-ID process.

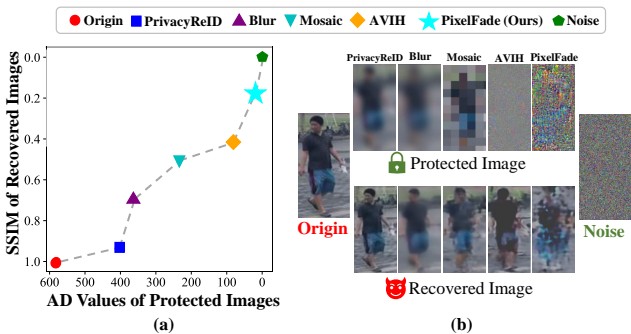

**Figure 1: (a) The potential influence of pixel distribution on resisting recovery attacks in protected images. An AD value (from Anderson-Darling [18] tests) close to zero signifies that the pixels of the protected image closely align with a normal distribution, signifying more chaos image pixels. Lower SSIM indicates lower quality of the recovered images, signifying stronger resistance to recovery attacks. (b) Visualization of protected and recovered images from different privacy-preserving person re-identification (PPPR) methods.**

However, potential data leakage [4] raised public concern because pedestrian images contain a large amount of personal information (*e.g.* facial information, profile, appearance, and texture). Public concerns motivate the development of the privacy-preserving person re-identification (PPPR) task [1, 19, 29, 30], which aims to protect the visual information of pedestrian images while maintaining their discriminative features for authorized models.

Existing PPPR methods can be roughly divided into two categories: First, conventional methods visually scramble the body of images via blurring, mosaic, or noise adding. Such methods injure the semantic features of the image, leading to a drop in Re-ID performance. Second, deep learning-based methods [19, 30] achieve a good balance between privacy and utility by transforming images into visually obfuscated images that can be recognized by the authorized Re-ID model. However, the above methods face the risk of recovery attacks [8, 15, 29, 32]. If malicious adversaries are aware of the principle of protection methods or have access to black-box control of the privacy model, they can launch recovery attacks by training a recovery network on the public dataset to *learn* the mapping from the protected image to the original image. Then the trained recovery network can reverse the protected image to the original image, leading to privacy leakage.

To deal with the above problem, we aim to make the protected image resistant to recovery attacks, while hiding their visual information and maintaining the utility for authorized Re-ID models. We start with the Normality Testing [18] on protected images from previous privacy-preserving methods to measure their pixel *chaos* degree. Here we measure the chaos degree by calculating the

similarity of the protected image to a normal distribution via the Anderson-Darling test [18], where lower values from the test (AD values) represent more chaotic protected images. As illustrated in Figure 1(a), the following phenomenon was observed: As the pixels of a protected image are *more chaotic*, the quality of the recovered image deteriorates, suggesting an increase in *resistance* to recovery attacks. We speculate that the *inherent randomness* of pixels of protected images can disrupt the recovery network's *learning* of the mapping from the privacy image to the original image, effectively diminishing the recovery capability of the adversary. Therefore, this inspires us to consider the privacy-preserving image recognition task from a new perspective: **Can a pedestrian image be converted into a nearly normal-distributed noise image to resist recovery attacks as well as protect visual privacy**?

However, naively converting images to random noise damages semantics information, leading to severe loss of discriminative features. It is a challenge to balance the trade-off between privacy and the utility of images. Fortunately, some works regarding adversarial attacks [5, 10, 16] show that deep neural networks (DNN) understand images in a different way from humans. In the TypeI adversarial attack [19, 20], the process transforms the image into a visually different one, but the model persists in recognizing it as belonging to the same identity. The above approach gives us a feasible way to preserve the high recognition performance of a Re-ID model for visually dissimilar images.

In this paper, we provide a simple yet effective method to iteratively optimize pedestrian images into noise-like images to perform PPPR tasks. We define our Noise-guided Objective Function as approximating pedestrian images to normal-distributed noise images to resist recovery attacks and protect privacy. During optimization, a feature constraint is imposed on the feature distance between protected and original images in the feature space of the pre-trained Re-ID model, thereby preserving the utility of protected images. However, Solving the above objective function is a non-convex optimization problem, simple optimization methods cannot find the local optimal point (refer to Section 4.4.2 for more analysis), which seriously impacts the privacy performance or Re-ID performance.

To achieve a good balance between privacy and utility, we propose a heuristics optimization strategy, named Progressive Pixel Fading, to process pixels by replacing them with random noise in a progressive manner. Specifically, we iteratively perform the Constraint Operation and the Partial Replacement Operation alternately according to the satisfaction of feature constraints. In the Constraint Operation, we follow TypeI Attack [19] to derive gradients to update protected images to minimize their feature loss with original images. In the Partial Replacement Operation, only part of scattered pixels are replaced with noise. Our Progressive Pixel Fading offers superior advantages in terms of both privacy and utility. On the one hand, the replacement ensures that pixel-level information from the original image is discarded to safeguard privacy. On the other hand, the unreplaced coarse-grained appearance (*e.g.* color, texture, and contour) of the pedestrians can effectively guide the optimization direction in Constraint Operation to facilitate the preservation of discriminative features.

We present a comprehensive experiment with our method (named PixelFade) on three widely used Re-ID datasets. Compared to previous PPPR methods, our PixelFade achieves the best results in

terms of resistance to recovery attacks and Re-ID performance. The visualization of protected images shows that PixelFade effectively protects the visual information of pedestrian images. Moreover, our PixelFade can be easily adapted to a multitude of Re-ID network architectures, and diverse Re-ID scenarios, highlighting its high scalability and applicability.

Our main contribution can be summarised as three-fold:

(1) Based on experimental findings, we introduce a Noise-guided Optimization Objective with feature constraints to optimize pedestrian images to protect visual privacy and resist recovery attacks.

(2) We propose Progressive Pixel Fading to replace pixels with noise progressively, aiming to efficiently retain the discriminative features within pedestrian images.

(3) Extensive experiments demonstrate our PixelFade outperforms state-of-the-art PPPR methods in terms of Re-ID performance and resistance performance.

## 2 RELATIVE WORK

### 2.1 Person Re-Identification

Person re-identification (Re-ID) aims to match individuals across different camera views or at different times within a surveillance network. With the development of deep learning, many works adopt or develop deep convolutional network architectures (*e.g.*, ResNet [6], MobileNet [9], OSnet [33]) to extract features from pedestrian images. Some works [7, 11] extract pedestrian features by developing the Transformer architecture [21]. To adapt to more practical scenarios, Text-to-Image Re-ID methods [11] aim to match textual descriptions of individuals with their corresponding images across different camera views, and Visible Infrared [17, 26, 27] Re-ID methods aim to address the challenge of Re-ID across visible light and infrared image modalities. To match pedestrians from different cameras, it is usually necessary to upload images and store them in the cloud. However, potential data leakage [4] can result in images being exposed to malicious attackers, potentially leading to tracking or even criminal incidents. To protect the privacy of pedestrian images, we propose a privacy-preserving method that preserves the discriminative features for Re-ID tasks.

### 2.2 Privacy-preserving Person Re-identification

Existing PPPR methods can be broadly categorized into two types. First, conventional approaches visually scramble the body in images through techniques such as blurring, mosaic, or adding noise. However, these methods compromise the semantic features of the image, resulting in decreased Re-ID performance. Second, deep learning-based methods [19, 30] achieve a good balance between privacy and utility. PrivacyReID [30] provides a joint learning reversible anonymization framework, capable of reversibly generating full-body anonymous images. AVIH [19] iteratively reduces the correlation information between the protected and original images to protect visual privacy while minimizing their distance in feature space. However, the above methods face the risk of recovery attacks [8, 15, 29, 32]. If malicious adversaries have access to black-box control of the privacy model, they can launch recovery attacks by training a recovery network on the public dataset to learn the mapping from the protected image to the original image. Recently,

Ye et al. [29] proposes Identity-Specific Encrypt-Decrypt architecture to encrypt the images to resist recovery attacks. However, the encrypted images cannot be used for retrieval by any Re-ID models. Our goal is to protect visual privacy of pedestrian images and resist recovery attacks while maintaining the performance of the authorized Re-ID model.

### 2.3 Adversarial Attacks

Many adversarial attack methods [5, 10, 16] show that deep neural networks (DNN) understand images in a different way from humans. In the TypeI adversarial attack [19, 20], the process iteratively transforms the image into a visually different one, but the model persists in recognizing it as belonging to the same identity. AVIH [19] strives to hide the visual information of face images while preserving their functional features for face recognition models. In our paper, we employ AVIH for PPPR task as a comparison method. In comparison, our method proposes a novel objective function that explicitly converts images to noise to resist recovery attacks and introduces a heuristic optimization strategy to effectively improve the privacy-utility trade-off.

## 3 PIXELFADE

In this section, we first introduce our Noise-guided Object Function with feature constraint to optimize pedestrian images to protected images in Section 3.1. Subsequently, we introduce our novel optimization strategy Progressive Pixel Fading in Section 3.2, followed by Constraint Operation in Section 3.3 and Partial Replacement Operation in Section 3.4.

### 3.1 Noise-guided Objective Function

Recall the experimental discovery in Section 1: As the pixels of a protected image are *more chaotic*, the quality of the recovered image deteriorates, suggesting an increase in *resistance* to recovery attacks. We speculate that the *inherent randomness* of pixels of protected images can disrupt the recovery network's *learning* of the mapping from the privacy image to the original image, effectively diminishing the recovery capability of the adversary. Therefore we take a novel perspective to tackle the PPPR task, with the explicit objective of converting pedestrian images to random noise to protect visual information and resist recovery attacks.

However, naively converting pedestrian images into noise images severely harms the semantic information and causes Re-ID performance to slip. We therefore introduce a feature constraint for limiting the feature distance between the protected image and the original image to be less than a preset threshold to maintain the Re-ID performance of the protected image.

Mathematically, we suppose there is a set of pedestrian images to be protected $X = \{x_1, \ldots, x_N\}$. For each $x_i$, we sample different noise images $\eta_i \sim \mathcal{N}$, where $\mathcal{N}$ is the standard normal distribution. Our objective function is defined as:

$$\min_{x_i^p} \left\| x_i^p - \eta_i \right\|_F^2,$$
$$\text{s.t } \left\| f\left(x_i^p\right) - f(x_i) \right\|_2^2 \leq \varepsilon, \tag{1}$$

where $x_i$ is $i$-th original image and $x_i^p$ is $i$-th protected image. $f$ is a pre-trained Re-ID model. For simplicity, we neglect the index $i$ of images in subsequent passages.

### 3.2 Progressive Pixel Fading

In this subsection, we aim to employ a suitable optimization strategy to optimize images into noise. Since Equation (1) is a non-convex optimization problem, simple optimization methods such as Random Perturb or L1 Optimization are not able to find the local optimum point (refer to Section 4.4.2 for more analysis), which seriously affects the Re-ID performance or privacy performance. To solve this problem, we propose a *heuristic* optimization strategy, named Progressive Pixel Fading, to update protected images.

The pipeline of PixelFade is depicted in Figure 2(a), where for a given pedestrian image $x$ requiring protection, we initially generate a random noise image $\eta$ sampled from $\mathcal{N}$ along with a set of binary masks $\mathcal{M}$. During optimizations, we iteratively carry out **Constraint Operation** (detailed in Section 3.3) to update the protected image to narrow the feature distance between the protected image and the original image, aiming to meet the feature constraint. If the feature distance is less than a specific threshold $\epsilon$, we proceed with one **Partial Replacement Operation** (detailed in Section 3.4) to replace parts of scattered pixels with noise to protect the privacy of protected images. Note that the replacement operation and the constraint operation are run *alternately* according to the satisfaction of feature constraints.

**Discussion.** We highlight the advantages of such heuristic optimization over simple optimization in terms of privacy and utility. For privacy, the replacement with noise values ensures that pixel-level information from the original image is *discarded* rather than merely *perturbed*, thereby safeguarding privacy. For utility, randomly masking out partially scattered pixels in the image drives the model to capture the intrinsic features from unmasked content, facilitating the preservation of discriminative features within the image during the next Constraint Operation. It helps improve the Re-ID performance of protected images.

### 3.3 Constraint Operation

To satisfy the feature constraints in Equation (1), we aim to minimize feature loss between protected and original images with the Type-I attack [20]. Specifically, we define optimization loss as the feature distance between the protected image and the original image for a particular Re-ID model. Formally,

$$\mathcal{L}_f\left(x_t^p, x\right) = \left\| f(x_t^p) - f\left(x\right) \right\|_2^2, \tag{2}$$

where $t$ indicates the step of optimization. Motivated by [3, 19], we calculate momentum gradients $g_t$ to stabilize optimization directions as:

$$g_{t+1} = \alpha \cdot g_t + \frac{\nabla \mathcal{L}_f\left(x_t^p, x\right)}{\left\| \nabla \mathcal{L}_f\left(x_t^p, x\right) \right\|_2}, \tag{3}$$

$$g_0 = \frac{\nabla \mathcal{L}_f\left(x_0^p, x\right)}{\left\| \nabla \mathcal{L}_f\left(x_0^p, x\right) \right\|_2}, \tag{4}$$

Figure 2: The framework of our PixelFade. Our goal is to optimize the original image $x$ towards the noise image $\eta$ to obtain the protected image $x^p$ for protecting visual information and resisting recovery attacks while retaining discriminative features. (a) The process of Progressive Pixel Fading. Constraint operation and Partial Replacement Operation are run alternately according to the satisfaction of feature constraints. (b) Partial Replacement Operation on the protected images. The randomly generated binary masks $\mathcal{M}_t^p$ are used to select the positions for replacing pixels with noise in the corresponding image.

where $\alpha$ indicates the decay factor of momentum computation. By applying backpropagation, we iteratively derive the gradient to update the protected pedestrian image to minimize its feature loss with the original image:

$$x_{t+1}^p = x_t^p - \beta \cdot g_{t+1} \tag{5}$$

## 3.4 Partial Replacement Operation

In this subsection, we describe the Partial Replacement Operation to protect privacy. In the person Re-ID task, pedestrian images contain a wealth of coarse-grained appearance including color, contour, texture, etc. The trained Re-ID model would consider such appearance information as important patterns of the pedestrian. Therefore our intuition is to leverage such coarse-grained appearance information as guidance to facilitate protected images to preserve discriminative features during the optimization in Constraint Operation.

In each Partial Replacement Operation, we employ a set of non-overlapping binary masks to replace part of the scattered pixels of the image with random noise. Specifically, we first preset a sequence of binary masks $\mathcal{M} = \{\mathcal{M}_1, \ldots, \mathcal{M}_I\}$, where $I$ denotes the number of masks. Each mask of different iteration $\mathcal{M}_j \in \{0, 1\}$ has the same shape as the original image. As shown in Figure 2(b), the template mask $\mathcal{M}'$ is entirely composed of the value of one. Then the masks for each iteration $\mathcal{M}_j$ are generated by randomly assigning a portion of pixels of $\mathcal{M}'$ to zero. Notice that each iteration does not select the previously picked pixels in one cycle. Formally, we replace pixels in the original images with noise via these masks:

$$x_{t+1}^p = x_t^p \odot \mathcal{M}_j + \mathcal{N} \odot (1 - \mathcal{M}_j). \tag{6}$$

Once the feature constraint is satisfied, one replacement operation is performed and $j = (j + 1) \mod I + 1$ is executed. Until the final mask $\mathcal{M}_I$, all pixels have been processed, guaranteeing the entire substitution of all pixels in the original image to safeguard privacy. Such a process leads to the *Fade* of pixels from the pedestrian image in a progressive manner, where the remaining

coarse-grained content would facilitate the model to obtain informative gradients in Equation (4) during backpropagation. It aids in updating the image in the Constraint Operation toward a more optimal direction, contributing to the preservation of discriminative features in the image.

---

**Algorithm 1:** PixelFade

**Input:** original image $x$; pretrained Re-ID model $f$; set of masks $\mathcal{M}$;
**Input:** maximum number of iterations $T$; number of masks $I$; threshold of Feature Constraint $\epsilon$;
**Output:** protected image $x_T^p$.

1   $x_0^P = x$; $g_0 = 0$;
2   Initialize index of masks $j = 0$;
3   Random initialize noise image $\eta \sim \mathcal{N}$;
4   **while** $t < T$ **do**
5     Compute $L_f(x_t^p, x)$ via Equation (2);
6     **if** $L_t^f \geq \epsilon$ **then**
7       Update $x^p$ by **Constraint Operation** via (Equations (3) to (5)) ;
8     **end**
9     **else**
10       Update $x^p$ by **Partial Replacement Operation** via (Equation (6)) ;
11       $j = (j + 1) \mod I + 1$ ;
12     **end**
13     $t = t + 1$;
14   **end**

---

## 3.5 Overview and Application of PixelFade

The algorithm is summarized in Algorithm 1. Empirically, for better privacy protection, we first perform an initialization stage of a few steps, performing Constraint Operation and Partial Replacement

Operation in turn and ensuring that all pixels have been replaced. We then officially perform our Progressive Pixel Fading, where Partial Replacement Operation and Constraint Operation are run *alternately* according to the satisfaction of feature constraints. Partial Replacement Operations are performed cyclically, meaning that if all pixels have been replaced once, a new round of replacement will continue to be performed.

After reaching the pre-set maximum number of optimization steps, protected images instead of original images are saved in the cloud for Re-ID tasks. By feeding query and gallery images from different cameras to the authorized Re-ID model, both unprotected and protected images from the same identity can be correctly matched by the Re-ID model. If the protected images stored in the cloud are leaked and fall prey to a malicious attempt at recovery attacks, our method can robustly prevent the recovery of visual information, underscoring its effectiveness in thwarting recovery attacks.

## 4 EXPERIMENTS

### 4.1 Experiments Settings

*4.1.1 Datasets.* Three widely used datasets are used for experiments: Market-1501 [31], MSMT17 [25] and CUHK03 [14]. The Market-1501 dataset consists of 32,668 annotated bounding boxes under six cameras. The MSMT17 dataset comprises of 4,101 identities and 126,441 bounding boxes taken by a 15-camera network. The CUHK03 dataset includes 1,467 identities and 14,097 detected bounding boxes. Besides, we adapt our PixelFade to Text-to-Image Re-ID on CUHK-PEDES [13] and ICFG-PEDES [2], to Visible-Infrared Re-ID on SYSU-MM01 [26] and RegDB [17] to demonstrate PixelFade's scalability.

*4.1.2 Threat Models.* We consider that the adversary can access black-box control of the privacy model and obtain protected images. Following existing recovery attacks [29], the adversary can obtain protected images as labels by feeding numerous original images from the public dataset (training set of Market-1501 or CUHK03) to the privacy model. Then adversary trains the recovery network to learn the mapping by minimizing the L1 loss between recovered and original images. After training, the adversary can reverse the original images from protected images by the trained recovery network.

*4.1.3 Evaluation Metrics.* For Re-ID performance, we use Cumulative Matching Characteristics (a.k.a., Rank-k matching accuracy) [22], mean Average Precision (mAP) [31], and a new metric mean inverse negative penalty (mINP) [28]. Higher above metrics represent higher utility of pedestrian images. For resistance to recovery attacks, we adopt two widely used metrics, *i.e.* PSNR and SSIM [24] to measure the similarity between recovered and original images. Specifically, a lower PSNR and SSIM indicate a lower similarity to original facial images, indicating better privacy protection.

*4.1.4 Implementation Details.* We follow the default training of AGW [28] on Re-ID datasets to obtain pre-trained Re-ID models. Unless specified, we use the ResNet50 [6] with non-local [23] block network as the backbone. We set the maximum number of iteration steps of PixelFade $T$ to 100 and the number of steps in the initialization phase is 10 out of 100 steps. The number of masks $\mathcal{I}$ is set

to 5. The threshold of Feature Constraint $\epsilon$ is 0.03. The decay factor $\alpha$ is 0.6. Note that we do not perform any replacement operation in the last 5 steps to ensure that the feature constraint is satisfied.

For compared methods, we pick five methods that protect the visual privacy of images while maintaining the performance of models: For (1) FaceBlur [1], We follow the default parameters in the article to detect and blur the face part. For (2) PrivacyReID [30], we follow their open-source code to reproduce that work. For (3) Gaussian blur and (4) Mosaic, we follow the default setting in [30] to set their radius to 12 and 24 respectively. For (5) AVIH, We use their open-source code and follow their default parameters (except the iteration step) to perform the PPPR task. For a fair comparison with our PixelFade, we set the maximum number of iteration steps for both two methods to 100.

### 4.2 Results of Person Re-Identification

We follow the relative work [30] to evaluate the Re-ID performance under four test settings with different queries and galleries, which represent four different scenarios: **Protected to Protected**: Both query and gallery sets are protected images. **Original to Protected**: Query sets are original images while gallery sets are protected images. **Protected to Original**: Query sets are protected images while gallery sets are original images. **Original to original**: Both query and gallery sets are original images. The "Upperbound" implies that an unprotected ReID model trained on the unprotected dataset performs Re-ID retrieval on the unprotected data.

Table 1 shows Re-ID performance results under different Re-ID settings. We can see that our PixelFade outperforms other privacy protection methods in all four settings. Our method almost approaches Upperbound, i.e., the differences in Rank1 are only 1.4%, 5.9%, and 4.2% on the three datasets even in the most challenging setting (Protected to Protected). It is worth noting that our method outperforms another iterative method (AVIH) on the mINP metric for all three datasets (*i.e.*, 9.2%, 3.6%, 7.6%). This is because our Progressive Pixel Fading drives the model to maintain the intrinsic features within protected pedestrian images, facilitating the identification of difficult samples across different viewpoints.

### 4.3 Results of Privacy Protection

Here we investigate PixelFade's protection of privacy, which we evaluate in two aspects: resistance to recovery attacks and visual protection.

*4.3.1 Resistance to Recovery Attacks.* We suppose that the adversary launches recovery attacks on protected images as discussed in Section 4.1.2. We compare PixelFace with previous methods to evaluate its resistance performance against recovery attacks on datasets of Market-1501 and CUHK03. As shown in the "Recovered images" part of Figure 3, PixelFade's recovered images (column f) are chaotic and almost impossible to recognize the original identity. On the contrary, recovered images of other methods (column b-e) fail to resist the recovery attacks. They still reveal some pedestrian contours, or even almost consistent with the original image. Table 2 shows the qualitative results of resistance to reconstruction attacks. We can see that our method reaches the lowest PSNR and SSIM, indicating PixelFade outperforms other protection methods on resistance performance against recovery attacks.

**Table 1: Evaluation of Re-ID Performance on three Re-ID datasets. Rank-1 accuracy(%), mAP(%), and mINP(%) are reported.**

| Privacy Settings | Methods | Market1501 | | | MSMT17 | | | CUHK03 | | |
|---|---|---|---|---|---|---|---|---|---|---|
| | | Rank1 | mAP | mINP | Rank1 | mAP | mINP | Rank1 | mAP | mINP |
| Protected to Protected | Mosaic | 64.3 | 43.4 | 13.0 | 10.6 | 5.7 | 0.7 | 8.8 | 9.9 | 5.3 |
| | Gaussian Blur | 67.3 | 44.2 | 13.7 | 15.2 | 7.2 | 0.8 | 8.2 | 10.7 | 6.9 |
| | PrivacyReID | 89.2 | 74.3 | 39.4 | 48.7 | 28.5 | 4.9 | 33.2 | 34.7 | 25.0 |
| | AVIH | 91.2 | 79.5 | 48.7 | 59.0 | 37.8 | 6.1 | 58.3 | 51.5 | 36.7 |
| | **PixelFade** | **94.2** | **85.2** | **58.1** | **62.7** | **43.1** | **9.7** | **63.1** | **58.5** | **44.3** |
| Original to Protected | Mosaic | 75.3 | 53.6 | 17.2 | 16.3 | 8.7 | 1.0 | 17.7 | 17.6 | 9.1 |
| | Gaussian Blur | 40.1 | 25.4 | 6.3 | 21.3 | 10.7 | 1.4 | 14.6 | 14.8 | 8.6 |
| | PrivacyReID | 88.2 | 72.0 | 37.0 | 51.1 | 29.7 | 5.2 | 39.2 | 38.4 | 27.2 |
| | AVIH | 92.6 | 81.3 | 50.2 | 60.1 | 41.5 | 8.9 | 60.2 | 54.1 | 39.0 |
| | **PixelFade** | **95.0** | **86.5** | **60.7** | **64.9** | **46.9** | **12.2** | **65.7** | **62.2** | **48.7** |
| Protected to Original | Mosaic | 70.9 | 54.7 | 24.1 | 14.6 | 9.0 | 1.6 | 15.1 | 17.7 | 12.3 |
| | Gaussian Blur | 18.3 | 15.5 | 5.2 | 16.2 | 9.4 | 1.5 | 10.4 | 12.4 | 8.4 |
| | PrivacyReID | 82.5 | 67.5 | 36.0 | 50.5 | 30.5 | 5.7 | 35.3 | 35.5 | 25.4 |
| | AVIH | 92.4 | 81.1 | 50.9 | 59.8 | 41.1 | 8.3 | 58.7 | 55.5 | 40.3 |
| | **PixelFade** | **94.3** | **86.4** | **61.7** | **63.1** | **46.4** | **11.6** | **63.4** | **61.1** | **49.1** |
| Original to Original | Mosaic | 87.4 | 73.4 | 39.5 | 25.0 | 15.3 | 2.6 | 28.5 | 31.5 | 22.9 |
| | Gaussian Blur | 84.8 | 67.4 | 32.2 | 30.5 | 17.1 | 2.8 | 30.4 | 31.5 | 22.3 |
| | PrivacyReID | 91.6 | 79.4 | 47.4 | 51.5 | 31.1 | 6.0 | 41.9 | 41.7 | 30.4 |
| | AVIH | 95.7 | 88.6 | 66.7 | 68.6 | 49.8 | 15.0 | 67.3 | 65.8 | 54.6 |
| | PixelFade | 95.7 | 88.6 | 66.7 | 68.6 | 49.8 | 15.0 | 67.3 | 65.8 | 54.6 |
| Unprotected (UpperBound) | | 95.7 | 88.6 | 66.7 | 68.6 | 49.8 | 15.0 | 67.3 | 65.8 | 54.6 |

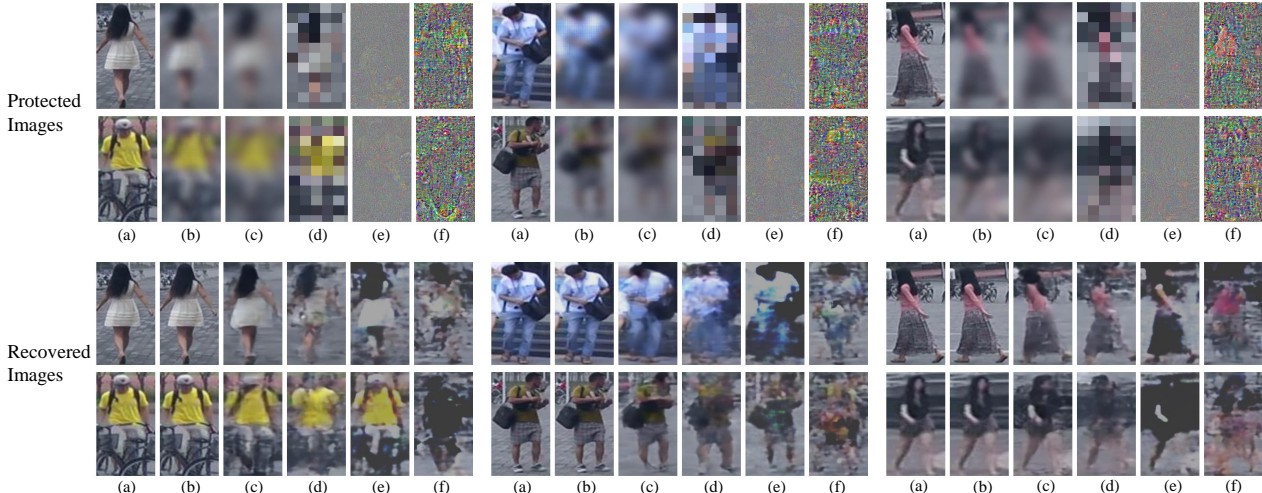

**Figure 3: Qualitative results of protected and recovered images from different privacy-preserving PPPR methods. (a) Origin; (b) PrivacyReID [30]; (c) Blurring; (d) Mosaic; (e) AVIH [19]; (f) Our PixelFade.**

*4.3.2 Visual Protection.* The "Protected images" part of fig. 3 shows the qualitative results, which visualize the protected images of different methods. Previous methods (columns b-d) still expose some visual information (*e.g.*, clothing color, contour). In comparison, our PixelFade (column f) effectively hides the visual information of pedestrians, which is almost consistent with noise images, making it difficult for malicious attackers to distinguish the identity.

## 4.4 Ablation Studies of PixelFade

In this subsection, we would like to demonstrate the superiority of our Noise-guided Objective Function and Progressive Pixel Fading through ablation experiments. All ablation studies are conducted on the Market1501 dataset.

*4.4.1 Noise-guided Objective Function.* First, we would like to verify the conjecture we presented in Section 1: As the pixels of the protected image become more chaotic, its ability to resist recovery

**Table 2: Quantitative results of resistance to recovery attacks. "PSNR" and "SSIM" indicates the quality of recovered images by malicious attackers. "AD" indicates the value of protected images from the Anderson-Darling test. The best is in bold.**

| Datasets | Methods | PSNR↓ | SSIM↓ | AD |
|----------|---------|-------|-------|-----|
| Market1501 | PrivacyReID | 26.92 | 0.94 | 401.29 |
|  | Gaussian blur | 23.24 | 0.69 | 363.63 |
|  | Mosaic | 17.76 | 0.51 | 232.14 |
|  | AVIH | 14.30 | 0.42 | 82.36 |
|  | **PixelFade** | **11.37** | **0.18** | **19.83** |
| CUHK03 | PrivacyReID | 23.94 | 0.89 | 352.15 |
|  | Gaussian blur | 20.12 | 0.64 | 289.01 |
|  | Mosaic | 17.12 | 0.48 | 194.88 |
|  | AVIH | 14.69 | 0.44 | 72.12 |
|  | **PixelFade** | **9.04** | **0.05** | **18.55** |

**Table 3: Analysis of the effect of pixel chaos degree on recovery attacks. For "Weight of Noise", we linearly interpolate the normal-distributed noise with the original image to varying degrees. "AD" is the value from the Anderson-Darling test, indicating the pixel chaos degree of protected images. "PSNR" and "SSIM" denotes the quality of recovered images.**

| Weights of Noise | AD | PSNR↓ | SSIM↓ | Rank1↑ | mAP↑ | mINP↑ |
|------------------|-----|-------|-------|--------|------|-------|
| 0.2 | 231.00 | 14.99 | 0.48 | 93.8 | 84.2 | 56.0 |
| 0.4 | 112.00 | 14.49 | 0.44 | 93.8 | 84.5 | 56.6 |
| 0.6 | 43.35 | 14.70 | 0.41 | 94.2 | 84.9 | 57.5 |
| 0.8 | 29.07 | 13.88 | 0.36 | 94.5 | 85.3 | 58.4 |
| 1.0 (Ours) | 19.83 | 10.92 | 0.18 | 94.2 | 85.2 | 58.1 |

**Table 4: Ablation Study of the objective function. The objective images are replaced with other images.**

| Objective | AD | PSNR↓ | SSIM↓ | Rank1↑ | mAP↑ |
|-----------|-----|-------|-------|--------|------|
| Images of Other Identity | 546.21 | 17.24 | 0.53 | 94.7 | 85.7 |
| Zero Images | 117.85 | 12.76 | 0.25 | 94.4 | 85.2 |
| Contrastive Images | 36.75 | 13.43 | 0.43 | 93.8 | 84.7 |
| Noise Image (Ours) | 19.83 | 10.92 | 0.18 | 94.2 | 85.2 |

attacks increases. We sample a noise image from the Gaussian distribution with the same shape as the pedestrian image, and then we mix it with the original image. We replace the objective images (*i.e.*, $\eta$ in Equation (1)) in our objective function in PixelFade with such a mixed noise image. The result is shown in Table 3. We can observe that as AD values decrease, implying that the pixel chaos degree in protected images is increasing, the quality of the restored image is deteriorating, indicating an increase in resistance to recovery attacks. This suggests that the random property of the pixels disrupts the learning of recovery networks, weakening the threat of recovery attacks. Therefore our PixelFade is dedicated to providing a new perspective to realize the privacy-preserving image recognition tasks that are transforming images into nearly normal-distributed noise images to resist recovery attacks.

To further evaluate the effectiveness of our objective function, we replace the objective images with other images instead of noise images. As shown in Table 4, when the objective images are "images of other identity", it achieves a high SSIM of 0.53 that it completely

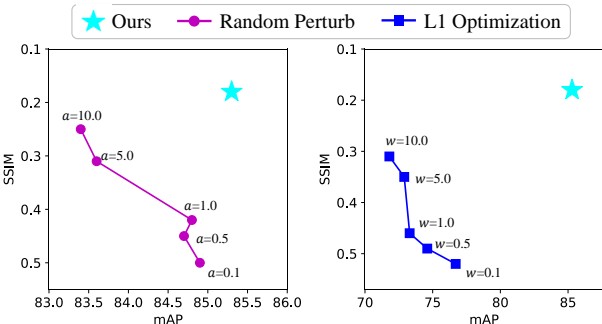

**Figure 4: Ablation study of optimization strategy. "Random Perturb" indicates adding random noise of magnitude $a$ to pedestrian images. For "L1 Optimization", we optimize the pedestrian image using L1 loss between noise images and original images, where loss is weighted as $w$.**

fails to resist recovery attacks, as natural images are poorly able to disrupt the learning of recovery networks. When the objective images are "Contrastive Images", our goal is to enlarge the difference between protected images and original images, which can be formally defined as:

$$\max_{x^p} \left\| x^p - x \right\|. \tag{7}$$

It achieves a higher PSNR of 2.51 and a higher SSIM of 0.25 compared to ours, indicating a lower resistance to attacks. If we set objective images as "Zero Images", we attempt to optimize protected images to be zero-valued images. It is still weaker than us in resisting recovery, with a difference of 1.84 in PSNR and 0.07 in SSIM. In comparison, our Noise-guided Objective Function explicitly optimizes the image into noise, which effectively disrupts the learning of recovery networks, promoting resistance to reconstruction attacks.

*4.4.2 Progressive Pixel Fading.* We employ other optimization strategies instead of our Progressive Pixel Fading to optimize images to noise shown in Figure 4. When "Random Perturb" is employed, we randomly generate noise with different amplitudes $a$, and add it to the protected image for perturbation. As shown in the left subplot of Figure 4, As the noise amplitude increases, the SSIM of the recovered image decreases, implying an increase in resistance to attacks. However, the accompanying side effect is that the Re-ID performance is also impaired. When "L1 Optimization" is employed, we minimize the L1 loss between protected images and noise images, where $w$ is the weight to balance L1 loss and Equation (2). From the right subplot of Figure 4 we can see that no matter what value of w is taken, the resistance performance and Re-ID performance are still far lower than ours. We suppose that the reason for the poor resistance of the above optimization strategies is that simple perturbation cannot completely remove the original information from pedestrian images. In comparison, our Progressive Pixel Fading completely discards the pixel-level information from the original image to ensure effective privacy protection. Meanwhile, the progressive way can motivate the model to effectively capture the intrinsic features of pedestrian images. The

Anonymous Authors

**Table 5: Results on Text-to-Image Re-ID scenario. We employ IRRA [12] method here for Baseline.**

| Datasets | Methods | Rank1↑ | mAP↑ | mINP↑ | PSNR↓ | SSIM↓ |
|---|---|---|---|---|---|---|
| CUHK-PEDES | IRRA w/ AVIH | 65.47 | 58.74 | 42.76 | 14.77 | 0.45 |
| | IRRA **w/ PixelFade** | **71.82** | **63.72** | **48.77** | **9.35** | **0.07** |
| | IRRA | 73.39 | 66.13 | 50.24 | +∞ | 1.00 |
| ICFG-PEDES | IRRA w/ AVIH | 39.29 | 38.73 | 27.25 | 14.89 | 0.38 |
| | IRRA **w/ PixelFade** | **45.63** | **45.26** | **33.08** | **10.31** | **0.13** |
| | IRRA | 47.24 | 47.52 | 35.04 | +∞ | 1.00 |

**Table 6: Results on Visible Infrared Re-ID scenario. We employ AGW [28] method here for Baseline.**

| Datasets | Methods | Rank1↑ | mAP↑ | mINP↑ | PSNR↓ | SSIM↓ |
|---|---|---|---|---|---|---|
| SYSU-MM01 | AGW w/ AVIH | 39.42 | 41.28 | 31.63 | 14.51 | 0.49 |
| | AGW **w/ PixelFade** | **43.74** | **44.94** | **33.72** | **9.34** | **0.11** |
| | AGW | 47.50 | 47.65 | 35.30 | +∞ | 1.00 |
| RegDB | AGW w/ AVIH | 63.25 | 59.24 | 41.76 | 15.31 | 0.51 |
| | AGW **w/ PixelFade** | **67.32** | **63.48** | **47.30** | **10.36** | **0.16** |
| | AGW | 70.05 | 66.37 | 50.19 | +∞ | 1.00 |

**Table 7: Scalability of PixelFade in terms of Re-ID network structure. We employ AGW [28] method here. Only the Re-ID performance of "Protected to Protected" scenario is shown.**

| Datasets | | Market1501 | | | MSMT17 | | |
|---|---|---|---|---|---|---|---|
| Re-ID BackBone | Protection | Rank1 | mAP | mINP | Rank1 | mAP | mINP |
| MobileNetV2 | w/ AVIH | 86.9 | 69.6 | 28.5 | 65.7 | 35.1 | 2.5 |
| | **w/ PixelFade** | **89.4** | **74.8** | **39.1** | **67.9** | **40.8** | **3.6** |
| | w/o Protection | 91.0 | 78.3 | 44.3 | 69.4 | 44.2 | 8.6 |
| OSNet | w/ AVIH | 91.5 | 79.6 | 48.6 | 77.4 | 51.2 | 8.6 |
| | **w/ PixelFade** | **93.1** | **83.2** | **55.1** | **79.9** | **56.7** | **11.2** |
| | w/o Protection | 94.8 | 86.9 | 62.8 | 81.2 | 60.6 | 17.1 |
| TransReID | w/ AVIH | 90.6 | 74.5 | 48.2 | 79.9 | 56.4 | 9.7 |
| | **w/ PixelFade** | **92.1** | **81.7** | **56.8** | **82.4** | **60.5** | **13.1** |
| | w/o Protection | 95.1 | 89.0 | 67.4 | 85.3 | 67.7 | 20.4 |

above advantages allow our optimization strategy to achieve the optimal trade-off between privacy and utility.

## 4.5 Scalability of PixelFade

A well-applied PPPR method should generalize to different scenarios and backbones. We transfer our PixelFade to other Re-ID scenarios, namely (1) Text-to-Image person Re-ID, aiming at searching protected images by text, and (2) Visible Infrared person Re-ID, which aims at searching the protected infrared image using the original RGB image. We choose an iterative method AVIH for comparison, and the experiment results are shown in Table 5 and Table 6. Our method outperforms AVIH in different scenarios with different datasets, and the gap to the upper bound is relatively small, suggesting the superior transferability of PixelFade in different Re-ID scenarios.

We then demonstrate the experiment of our PixelFade's generalization to different backbones in Table 7. We selected three commonly used Re-ID backbones for experiments, which have similarly strong Re-ID performance on Market1501 and MSMT17 datasets under the "Protected to Protected" setting. The above experiments demonstrate the high scalability and practicality of our method.

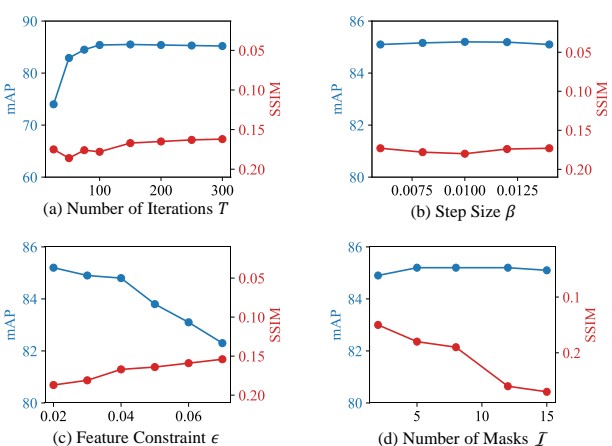

**Figure 5: Parameter analysis of PixelFade. Larger mAP indicates higher Re-ID performance. Smaller SSIM means stronger privacy performance.**

## 4.6 Parameter Analysis of PixelFade.

In this subsection, we provide an analysis of the impact of some critical parameters in PixelFade on privacy performance and Re-ID performance as shown in Figure 5. All experiments of parameter analysis are conducted on the Market1501 dataset.

Figure 5(a) shows the convergence of our PixelFade on both ReID performance and privacy performance. As the number of iterations increases, the Re-ID performance rises and the SSIM decreases. After 100 steps, the two metrics are almost constant, implying that our optimization reaches convergence in both two tasks. Figure 5(b) verifies the robustness of the choice of step size $\beta$. The default $\beta$ is 0.01, and the result implies that a beta in the range of $0.01 \pm 0.005$ is robust. Figure 5(c) demonstrates the influence of different feature thresholds $\epsilon$ on the results. As the $\epsilon$ increases, which means that the distance between protected and original images becomes farther, leading to a decrease in Re-ID performance and an increase in privacy performance. PixelFade is robust to the feature threshold $\epsilon$ when it is less than 0.04. Figure 5(d) shows the influence of different number of masks $\mathcal{I}$ on the results. Larger $\mathcal{I}$ means sparser pixels are replaced in each Partial Replacement Operation, which offers more remaining information to prompt the model for better optimization. When $\mathcal{I}$ is in the range from 2 to 8, the privacy performance and Re-ID performance are stable. Generally, our PixelFade is robust to parameter selection.

## 5 CONCLUSION

In this paper, we propose an iterative method to explicitly optimize pedestrian images into noise-like images to resist recovery attacks while maintaining Re-ID performance for authorized Re-ID models. Extensive experiments demonstrate the superior performance of our PixelFade in resisting recovery attacks and Re-ID performance compared to previous methods. Moreover, we experimentally show that our PixelFade can be easily adapted to diverse Re-ID scenarios and network backbones, highlighting its practicality and applicability.

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
