# OpenReview forum: "PixelFade: Privacy-preserving Person Re-identification with Noise-guided Progressive Replacement"
_acmmm.org/ACMMM/2024/Conference — MM2024 Poster_

### Official Review · Reviewer_2ZTL · 2024-05-19

**Rating:** 3
**Confidence:** 3

**Summary:**

This paper defines a Progressive Pixel Fading scenario to optimize pedestrian images into noise-like images to resist recovery attacks for person re-identification. Specifically, the approach introduces a Noise-guided Optimization Objective with feature constraints to protect visual privacy and resist recovery attacks. Experimental results on several benchmark datasets are reported.

**Strengths:**

1. The language is concise and easy to understand.
2. The experiments and ablation studies are extensive.

**Limitations:**

Some concerns and suggestions to improve this work are as follows:
1. The abstract is too long and could benefit from being shortened. I suggest revising the abstract to condense the information and focus on the most essential aspects of the research.
2. The main technology of this paper comes from AVIH [19]. I think the main contribution is Partial Replacement Operation, but the premise of this operation has not been given theoretical or experimental proof, and it is difficult to ensure its authenticity.
3. The model architecture could be included. Especially, the explanation of the architecture is not obvious in Figure 2.
4. Can the model handle any privacy protection in the different pedestrian images? Will any separate Noise-guided Progressive Replacement be useful?
5. The language expression in the manuscript needs improvement. I strongly suggest polishing for clarity and coherence.

**Suitability:**

2

---

### Official Review · Reviewer_kHKB · 2024-05-21

**Rating:** 5
**Confidence:** 3

**Summary:**

The paper studies the pedestrian privacy protection method. To this end, they propose an iterative method to explicitly optimize pedestrian images into noise-like images to resist recovery attacks while maintaining Re-ID performance for authorized Re-ID models.

**Strengths:**

The authors propose PixelFade to optimize pedestrian images into noise-like images to resist recovery attacks. Abundant experiments prove its effectiveness on different tasks.

**Limitations:**

1、	The authors propose PixelFade to resist recovery attacks. However, there are many ways of image recovery. How to implement image recovery in experimental comparison? Different methods of recovery may have different results.
2、	The bigger the noise, the harder it is to recover. In Table 3, the author conducted experiments. What is the effect if the noise is becoming larger? Is it more difficult to recover? What is the impact on the recognition effect?
3、All comparison methods should have the reference in all tables.

**Suitability:**

3

---

### Official Review · Reviewer_hYmX · 2024-05-24

**Rating:** 3
**Confidence:** 3

**Summary:**

This paper focus on the face privacy problem in person re-identification scenario. The authors proposed an iterative method that change the original images into noise-like ones, aims to preserve face privacy and avoid impacting re-identification method too much.

**Strengths:**

1) Privacy is a practical problem to be solved, especially in visual recognition areas.
2) From the results, the proposed method seems to be effective.
3) The authors promise to release the source code for reproducibility.

**Limitations:**

Although the proposed method seems to be effective from the current results, I think the evaluations are not enough.
1) Only one Re-ID method is adopted for evaluating the influence on Re-ID methods (i.e., results in Table 1). More Re-ID methods should be involved to check the effectiveness.
2) For the recovery attacks, I suggest applying the Re-ID methods to the recovered images and presenting the results, which will be more valid than image quality metrics. Otherwise, there is a possibility that although the recovered images with low image quality are not recognizable for humans, they are easier to be recognized for Re-ID methods.

**Suitability:**

2

---

### Official Review · Reviewer_M5cL · 2024-05-24

**Rating:** 6
**Confidence:** 4

**Summary:**

This paper proposes PixelFade, a novel method for privacy-preserving person re-identification (PPPR) that optimizes pedestrian images into noise-like images. The key idea is to resist recovery attacks while preserving discriminative features for authorized re-id models. The authors introduce a Noise-guided Objective Function with feature constraints to guide the optimization process. They also propose a Progressive Pixel Fading strategy, which alternates between a Constraint Operation to maintain re-identification performance and a Partial Replacement Operation to replace pixels with noise for privacy protection. Experiments on multiple datasets demonstrate the work's superior performance in terms of resistance to recovery attacks and re-identification accuracy compared to state-of-the-art methods.

**Strengths:**

- Progressive Pixel Fading strategy effectively balances privacy and utility by alternating between constraint and replacement operations
- Extensive experiments demonstrating superior performance in resisting recovery attacks and maintaining re-identification accuracy
- Scalability to different re-identification scenarios (e.g., text-to-image, visible-infrared) and network architectures
- Comprehensive ablation studies and parameter analysis provide insights into the method's effectiveness and robustness
-  The reviewer likes the idea that converting normal images into noise-like representations but preserving the the person re-id model's functionality because it also solves the visual privacy concerns. The idea is similar to https://dl.acm.org/doi/10.1145/3524273.3528189 but this work focuses on the image classification only.

**Limitations:**

The reviewer would like to see the discussions on the potential failure cases of the proposed work (or the successful rate). The computation resources and timing performance during the optimization process could be further analyzed and discussed.

**Suitability:**

3

---

### Meta-Review · Area_Chair_Jmxk · 2024-07-04

**Recommendation:** Accept (Poster)
**Confidence:** 4

**Metareview:**

This paper defines a Progressive Pixel Fading scenario to optimize pedestrian images into noise-like images to resist recovery attacks for person re-identification. Pre-rebuttal this paper received diverse ratings, i.e. 1 A, 2 BR, 1 WA. The main concerns were only one Re-ID method for evaluation (R2), missing comparison of different methods of image recovery (R3), and the main contribution is not theoretically or experimentally verified(R4). Post-rebuttal the paper still had diverse ratings: 1 A, 1 WR and 2 BA. Two reviewers downgraded the ratings. The rebuttal addressed some of the concerns. However, concerns remained about no numeric evidence of the main contributions (R4).
After careful consideration of the rebuttal, AC agrees with R1, R2 and R3’s comments that this paper is dealing with a practical problem and proposing an effective strategy. The remaining issues of reviewers are mainly about more details or experiments, which are difficult to be fully addressed in the rebuttal. However, these issues should be fully considered and addressed in the camera ready. AC tends to accept this paper.

---

### Meta-Review · Senior_Area_Chairs · 2024-07-10

**Recommendation:** Accept (Poster)
**Confidence:** 4

**Metareview:**

This paper received mixed ratings initially. After rebuttal, the final ratings are also diverse. Three tend to accept the paper and one tends to reject. AC carefully read the reviews and rebuttal and recommend accptence of the paper.SAC agrees with AC.